# Comparison of Two Diet and Exercise Approaches on Weight Loss and Health Outcomes in Obese Women

**DOI:** 10.3390/ijerph19084877

**Published:** 2022-04-17

**Authors:** Brittanie Lockard, Michelle Mardock, Jonathan M. Oliver, Mike Byrd, Sunday Simbo, Andrew R. Jagim, Julie Kresta, Claire C. Baetge, Yanghoon Peter Jung, Majid S. Koozehchian, Deepesh Khanna, Chris Rasmussen, Richard B. Kreider

**Affiliations:** 1School of Nursing and Health Professions, University of the Incarnate Word, San Antonio, TX 78209, USA; lockard@uiwtx.edu; 2Exercise & Sport Nutrition Lab, Human Clinical Research Facility, Department of Health and Kinesiology, Texas A & M University, College Station, TX 77843, USA; mamardock@gmail.com (M.M.); jo2667@tc.columbia.edu (J.M.O.); mbyrd@byrdseye.net (M.B.); sy.simbo@ctral.org (S.S.); ccbaetge@tamu.edu (C.C.B.); crasmussen@tamu.edu (C.R.); 3Byrd’s Eye Enterprises, Inc., Forney, TX 75126, USA; 4Center for Translational Research in Aging & Longevity, Human Clinical Research Facility, Department of Health & Kinesiology, Texas A&M University, College Station, TX 77843, USA; 5Department of Sports Medicine, Mayo Clinic Health System, Onalaska, WI 54650, USA; jagim.andrew@mayo.edu; 6College of Education and Human Development, Texas A&M University Central-Texas, Killeen, TX 76549, USA; jkresta@tamuct.edu; 7CJ CheilJedang, Suwon 16495, Korea; drpeter@cj.net; 8Department of Kinesiology, Jacksonville State University, Jacksonville, AL 36265, USA; mkoozehchian@jsu.edu; 9Department of Foundational Sciences, Nova Southeastern University, Clearwater, FL 33759, USA; drdeepesh@gmail.com

**Keywords:** body composition, fat loss, high-protein diet, training adaptations

## Abstract

Aim: To compare the efficacy of two popular weight loss approaches on weight loss, body composition, and markers of health in sedentary obese women. Methods: In total, 51 sedentary women (age 34.5 ± 7.7 yrs.; weight 90.0 ± 14.5 kg; BMI 34.0 ± 5.1 kg/m^2^; 46.5 ± 7.0% fat) were matched and randomized to participate in the Weight Watchers^®^ Momentum™ (WW) or Curves^®^ (CV) Fitness and Weight Management program for 16 weeks. Participants in the WW group (*n* = 27) were provided a point-based diet program, received weekly progress checks and counseling, and were encouraged to exercise. Participants in the CV group (*n* = 24) followed a menu-based higher protein/low-fat diet (1200 kcal/d) for 1 week; 1500 kcal/d diet for 3 weeks; and 2000–2500 kcals/d for 2 weeks that was repeated three times (except the last segment) while participating in a supervised circuit-style resistance training program (3 d/wk). A general linear model (GLM) with repeated measures was used to analyze data and are presented as mean changes from baseline (mean [UL, LL]). Results: Supervised CV training resulted in greater amounts of vigorous and total physical activity. After 16 weeks, both groups lost weight (WW −6.1 [−7.8, −4.6], CV −4.9 [−6.2, −3.2] kg, *p* = 0.264). Participants in the CV group observed greater reductions in fat mass (WW −2.9 [−6.7, −0.2], CV −6.4 [−9.2, −3.6] kg, *p* = 0.081) and increases in lean mass (WW −2.5 [−4.3, −0.7], CV 1.3 [−0.6, 3.2] kg, *p* = 0.005) resulting in more favorable changes in percent body fat (WW −1.4 [−4.1, 1.2], CV −4.7 [−7.5, −1.8]%, *p* = 0.098). Both groups observed improvements in peak aerobic capacity and muscular endurance, although bench press lifting volume was greater in the CV group. Those in the CV group experienced a greater increase in HDLc and reduction in the CHL–HDLc ratio and triglycerides. Conclusion: Both interventions promoted weight loss and improvements in fitness and markers of health. The CV program, which included supervised resistance training and higher protein diet menus, promoted greater fat loss, increases in lean mass, and improvements in percent body fat and blood lipids. Trial Registration: clinicaltrials.gov, #NCT04372771, registered retrospectively 1 May 2020.

## 1. Introduction

According to the 2013–2016 NHANES Survey, 49% of U.S. adults reported attempting to lose weight; this percentage grew to 56.4% when assessing women specifically. The most-reported weight loss attempts were to “exercise” and “eat less”, often in combination [1], which on the surface is a logical strategy, yet the success rates suggest otherwise. Traditionally, most weight-loss interventions have recommended reducing caloric intake in some manner (e.g., adherence to meal plans, monitoring caloric intake, use of meal replacements, provision of portion-controlled meals) and increasing low- to moderate-intensity aerobic exercise (e.g., walking, stationary cycling). Additionally, many weight loss programs incorporate behavioral strategies to improve adherence and efficacy (e.g., recording food intake, monitoring body weight, counseling, avoidance of energy-dense foods) [2,3,4,5]. Studies generally indicate that energy restriction plays a more important role in promoting initial weight loss and that exercise plays an important role in effectively maintaining weight loss and improving body composition parameters [6,7]. However, one of the challenges with diet-based weight loss interventions is a reduction in resting energy expenditure, which may be a byproduct of reductions in body mass, but more specifically reductions in fat-free mass, which may increase the likelihood of weight regain [8,9]. An abundance of research from our lab has reported effective weight and fat loss while also preserving fat-free mass and resting energy expenditure by utilizing a hypoenergetic high-protein diet combined with a supervised circuit-style resistance program in both pre- and post-menopausal women [8,9,10]. We have also shown improvements in markers of health and fitness with this protocol [11,12,13,14,15,16,17,18,19,20,21,22]. 

While caloric restriction underpins reductions in body mass, dietary alteration in macronutrients likely plays a role in facilitating favorable improvements in body composition parameters, particularly when combined with a structured resistance-training program. Specifically, an increase protein intake paired with resistance training may help with a greater preservation of fat-free mass during a weight loss program, when compared to caloric restriction alone; thereby potentially serving as a more effective strategy for long-term weight loss maintenance. Therefore, the purpose of this study was to determine whether adherence to a menu-based higher protein and low-fat diet with supervised resistance-based exercise (Curves^®^ Complete fitness and weight management program) is more effective in promoting fat loss than following a point-based diet program with weekly counseling and encouragement to exercise (Weight Watchers^®^ Momentum™). We hypothesized that participating in a supervised exercise program that included resistance exercise while consuming a higher protein/low-fat hypoenergetic diet would promote fat loss and the maintenance of fat-free mass to a greater degree than a more traditional diet intervention that included counseling and recommendations to increase physical activity. Primary outcomes included body weight, body composition, and resting energy expenditure. Aerobic capacity, muscular strength, muscular endurance, resting hemodynamics, fasting glucose homeostasis, and lipid-related variables were secondary outcomes.

## 2. Methods

### 2.1. Experimental Design

This study utilized a randomized, parallel, prospective intervention within a university-based research setting. This research protocol was approved by the Texas A&M Human Participant Protection Board (IRB2010-0130F) in accordance with the Declaration of Helsinki [23] and registered with clinicaltrials.org (#NCT04372771). Data were collected before and after 4, 6, 10, 12, and 16 weeks of intervention (see Figure 1).

### 2.2. Participants

Recruitment included advertisements in local newspapers, campus flyers, internet advertisements, and area physicians. Interested participants were pre-screened with a telephone interview. General entrance criteria included: (1) being an apparently healthy pre-menopausal female between the ages of 18 and 50 years with a body mass index (BMI) greater than 25 kg/m^2^, (2) not recently participating in an exercise or diet program, and (3) being capable of engaging in exercise training. Those who met the initial qualifications were invited to a familiarization session to receive an explanation of the study, sign consent forms, and complete personal and medical history information. After that, participants were randomly assigned to either: (1) Curves^®^ fitness and weight management program [24] (CV) or (2) Weight Watchers^®^ Momentum™ Program [25] with weekly meetings and encouraged exercise engagement (WW). Figure 2 presents a CONSORT diagram. A total of 149 women responded to research advertisements, 127 women met initial phone screening criteria, and 120 women underwent familiarization. Of these, a total of 98 women consented to participate in the study and 51 women (34.0 ± 6.1 years old, 90.1 ± 14.5 kg, 34.0 ± 5.1 kg/m^2^, and 46.5 ± 7.1% body fat) completed the study. Time constraint was the primary cause for those who withdrew (*n* = 22). Study recruitment and completion spanned approximately 24 months.

### 2.3. Diet Intervention

Participants were matched in accord with body mass and BMI, and then randomized into either the CV or WW interventions, in a similar fashion to other studies from our lab [10,12,13,15,16,17,19]. Those in the CV group followed the Curves^®^ Weight Management program as summarized in Figure 3. Participants were instructed to adhere to one-month weight loss cycles that involved consuming 1200 kcals/d for seven days during Phase I and 1500 kcals/d for 21 days during Phase II of the diet plan. During these diet phases, participants were provided meal plans developed by registered dietitians with macronutrient distributions consisting of 30% carbohydrate, 45% protein, and 25% fat. Next, the participants completed a metabolic recovery period (Phase III) in which they were allowed to ingest 2000–2500 kcals/d consisting of 45% carbohydrate, 30% protein, and 25% fat. During Phase III, participants were instructed to monitor body weight and reduce energy intake to 1200 kcals/d for 2–3 days if they experienced a three-pound or greater weight gain. Participants repeated this weight loss cycle three times with the recovery cycle twice, in between during the 16-week intervention. A registered dietitian reviewed the diet and exercise plans with participants prior to the start of the study and monitored compliance weekly to discuss any dietary challenges or concerns. Participants were encouraged to consume a multivitamin with 800 mg of calcium and 520 mg of Omega-3 fatty acids daily during the intervention.

Participants assigned to the WW intervention group were registered for the monthly subscription to the Weight Watchers^®^ Momentum™ Program [25]. Participants were required to attend one meeting each week with an 80% compliance record (13 out of 16 meetings) for inclusion in the final analysis. All participants attended the same Weight Watchers^®^ franchise location so that compliance could be monitored. Additionally, participants emailed the study coordinator each time they attended a meeting. Meetings consisted of an individual weigh-in, counseling by site personnel, and group discussions. Program materials were provided during the study and included provision of food plans using a point system, physical activity recommendations, and counseling related to managing feelings of hunger, successful dieting habits, strategies for eating away from home, how to track food intake and physical activity, managing eating with others, food recipes, how to vary exercise sessions, and dealing with weight plateaus. The program also encouraged participants to perform 30 min of physical activity on most days of the week.

### 2.4. Exercise Intervention

Participants in the CV intervention engaged in 30 min of supervised whole-body circuit training resistance exercise program three days/week. The CurvesSmart™ system was utilized for circuit program (Curves International, Waco, TX, USA) which was equipped with computerized software designed by MYTRAK (version 4.2.0.0, copyright 2004–2010, MYTRAK Health System, Mississauga, ON, Canada). The circuit included 13 bi-directional hydraulic resistance exercise machines which emphasized all major muscle groups in the concentric-only direction as previously described [18]. Participants were encouraged to perform as many repetitions within a 30 s period of time as possible on each resistance machine during each session. Between each machine, participants performed floor-based exercises (e.g., stepping, calisthenics) for 30 s to maintain an elevated heart rate. Thus, each participant performed 30 s on a machine, followed by 30 s of floor-based calisthenics until they completed two complete circuits (26 stations in total). Two sessions per week included Zumba^®^ in which participants performed one-minute periods on the resistance exercise machines followed by one-minute intervals of Zumba^®^ dance moves taught by a certified instructor. Upon completion of the exercise circuit, participants performed a brief cool down of stretching. Our lab has previously reported that this workout produces resistance exercise intensities ranging between 61–82% of 1RM, an exercise intensity of 65 ± 10% of peak oxygen uptake, a heart rate average of 80% HR_max_ (126 ± 15 bpm) and expends approximately 314 ± 102 kcals per workout [26,27,28]. Curves workouts were monitored for intensity and proper exercise technique by trained fitness instructors. Participants were considered compliant to the exercise intervention if they completed a minimum of 80% of the training sessions (38/48). Participants were also encouraged to walk for 30 min at a brisk pace on most days of the week at an exercise intensity of 60–80% of heart rate reserve. Participants in the WW intervention were encouraged to perform 30 min of physical activity on most days of the week.

## 3. Procedures

### 3.1. Dietary Assessment

Subjects met with a registered dietitian to learn accurate food intake documentation and portion size estimation. Participants documented 4 days (including one weekend day) of all food and fluid intake prior to each testing session at weeks 0, 4, 6, 10, 12, and 16. During each testing session, participant documentation was reviewed to ensure legibility, accuracy, and completeness. The Food Processor Version 9.1.0 (ESHA Nutrition Research, Salem, OR, USA) Nutrition Analysis Software was used to analyze average energy intake and macronutrient content. Dietary records were reviewed by a registered dietitian. 

### 3.2. Physical Activity Assessment

The International Physical Activity Questionnaire (IPAQ; 7 d version) was utilized to quantify physical activity patterns [29,30,31]. This questionnaire assessed the frequency and intensity of various types of physical activity including those related to occupational, transportation, housework, family-care, and recreation, sports, and leisure time. Metabolic equivalent levels (MET) were identified based on each activity’s intensity, with light (walking level) as 3.3 METs, moderate as 4.0 METs, and vigorous as 8.0 METs. The IPAQ has been validated as a tool to indicate physical activity pattern changes [29,30,31,32].

### 3.3. Resting Energy Expenditure and Metabolism

The ParvoMedics TrueMax 2400 Metabolic Measurement System (ParvoMedics, Inc., Sandy, UT, USA) was used to assess resting energy expenditure (REE). Participants laid down in a supine position on a comfortable bed and rested without sleeping for 20 min. The flow of air was controlled by a dilution pump to ensure a consistent flow of carbon dioxide (0.8–1.2%) through the metabolic cart. Measurements were recorded after the first 10 min of testing, during a five-minute period in which O_2_ uptake and CO_2_ expiration were within a 5% range. Previous results from our lab assessed the relative standard deviation for REE to range from 8.2–12.0%, with a mean intra-class correlation of 0.942 [12].

### 3.4. Body Composition and Anthropometrics

An electronic scale (Cardinal Detecto Scale Model 8430, Webb City, MO, USA) was calibrated with a precision of ±0.02 kg and utilized to determine body mass and height. The Gulick tensiometer tape measure was utilized to measure waist and hip circumference [33]. The Hologic Discovery W (Hologic Inc., Waltham, MA, USA) DXA equipped with APEX Software (APEX Corporation Software, Pittsburg, PA, USA) was used to measure body composition (excluding cranium). A previous study from our lab utilizing this DXA yielded test–retest reliability coefficients of variation of 0.31–0.45% for bone mineral content and lean mass, with an average intra-class correlation of 0.985 [15]. 

### 3.5. Resting Hemodynamics and Exercise Capacity

The radial artery was manually palpated to determine resting heart rate. Resting blood pressure was determined in the supine position after resting for about 5 min using a mercurial sphygmomanometer (American Diagnostic Corporation, model #AD-720, Hauppauge, NY, USA) according to standard clinical procedures [33]. Participants were attached to a Nasiff Cardio Card electrocardiograph (Nasiff Associates, Inc., Central Square, NY, USA) using a 12-lead arrangement according to standard procedures [33]. Participants then performed a standard Bruce protocol treadmill test on a Trackmaster TMX425C treadmill (JAS Fitness Systems, Newton, KS, USA) with a Parvo Medics 2400 TrueMax Metabolic Measurement System (ParvoMedics, Inc., Sandy, UT, USA). Isotonic 1RM bench press testing was measured on a standard Olympic bench press (Nebula Fitness, Versailles, OH, USA) using standard procedures with 2 min recovery between attempts [33]. Following a 5 min rest, upper body muscular endurance was determined by performing a maximum number of repetitions at 80% of 1RM. After another 5 min rest, participants performed lower body 1RM maximal strength on a hip sled/leg press (Nebula Fitness, Versailles, OH, USA) using a similar protocol as mentioned above [33]. Following a 5 min rest, lower body muscular endurance was determined by performing a maximum number of repetitions at 80% of 1RM. To calculate total lifting load-volume, the amount of weight in kg lifted was multiplied by the number of repetitions in the muscular endurance test. Previously, these strength tests yielded low mean coefficients of variation (bench 1.9%, hip sled 0.7%) and high test-to-retest reliability (intra-class bench r = 0.94, hip sled r = 0.91) in our lab [34].

### 3.6. Blood Collection and Analysis

Standard phlebotomy techniques were utilized to collect fasting whole blood and serum samples. Complete blood counts with platelet differentials were analyzed from whole blood using an Abbott Cell Dyn 3500 analyzer (Abbott Laboratories, Abbott Park, IL, USA). A complete metabolic panel was run on serum samples by Quest Diagnostics (Quest Diagnostics, 5850 Rogerdale Road, Houston TX, USA 77072) using the Olympus AAU 5400 Chemistry Immuno Analyzer (Olympus America Inc., Center Valley, PA, USA). Samples were re-run if the values exceeded normal ranges. Test–retest reliability of performing assays using this system range from 2% to 6%. Serum leptin and insulin were determined in duplicate using commercially available enzyme-linked immunosorbent assay (ELISA) kits (11-LEPHU-E01 and 80-INSHU-E10, ALPCO, Salem, MA, USA) utilizing standard procedures. Optical densities were determined at a setting of 450 nm using a BioTek ELX-808 Ultramicroplate reader (BioTek Instruments Inc, Winooski, VT, USA) and compared to a standard curve using BioTek Gen5 Analysis software (BioTek Instruments Inc., Winooski, VT, USA). The coefficients of variation (Cv) were as follows: intra-assay leptin 3.7 to 5.5% and insulin 2.9 to 6.2%, and inter-assay leptin 5.8 to 6.8% and insulin 5.4 to 8.6%. The homeostasis model assessment for insulin resistance (HOMA-IR) was used to estimate changes in insulin resistance using standard methods [35].

### 3.7. Statistical Analysis

Previous research from our lab utilizing similar diet and exercise interventions was utilized to set the a priori power calculation (>0.80), based upon the observed change in fat mass between diet groups [12,13,15,16,17,21], and revealed that to detect meaningful changes in fat mass (~2 kg) a sample size of 15–20 per group was needed. Those who completed at least 80% of the training sessions were included in the analysis. Missing data were replaced using the last observed value carry forward method value to represent individual responses. IBM^®^ SPSS^®^ version 28 Statistics for Windows (IBM Corp., Armonk, NY, USA) was used for analysis. Differences between groups at baseline was determined using general linear modal (GLM) multivariate analysis. Related variables were analyzed using GLM univariate, multivariate and repeated measures. *p*-levels are reported for overall multivariate Wilks’ Lambda and Greenhouse-Geisser univariate. Significance was set with a probability of type I error as 0.05 or less, while *p*-levels ranging between *p* > 0.05 to *p* < 0.10 were considered a statistical trend toward significance. Partial eta squared effect sizes (η_p_^2^) are reported as an indicator of effect size [36]. An eta squared value of 0.02 was considered small, 0.13 medium, and 0.26 large [36]. Differences among groups were determined using Tukey’s least significant differences (LSD) post hoc analyses. One-way ANOVA was utilized to analyze mean changes and percent changes from baseline with 95% confidence intervals (CI). Mean changes were considered significantly different when the 95% CI was completely above or below baseline [36]. Data are reported as means ± standard deviations, mean (upper limit [UL], lower limit [LL]) change from baseline, or mean (UL, LL) percent change from baseline. 

## 4. Results

### 4.1. Baseline Characteristics

Baseline demographics are presented in Table 1. Participants (*n* = 51) were 34.5 ± 7.7 years; 162.8 ± 7.0 cm; 90 ± 14.5 kg; 34.0 ± 5.1 kg/m^2^; and 46.5 ± 7.0% fat. No significant differences were observed between groups at baseline. 

### 4.2. Energy Intake and Resting Energy Expenditure

Appendix A presents self-reported energy and macronutrient intake as well as resting energy expenditure data. Significant Wilks’ Lambda time (*p* < 0.001) and group × time interaction (*p =* 0.006) effects were revealed with overall GLM multivariate analysis. Univariate analysis revealed significant group x time interactions in energy intake (*p* = 0.05), protein intake (*p* = 0.001), and fat intake (*p =* 0.002). Pairwise comparisons revealed that energy intake, carbohydrate intake, and fat intake decreased over time, with participants in the CV intervention generally consuming less carbohydrate and more protein than those following the WW diet. Relative resting energy expenditure increased over time in both groups, with no differences observed between groups. Present findings are consistent with prior reports from our lab, indicating that adherence to these diets promote significant reductions in energy and macronutrient intake, although participants generally consume lower than the planned energy intake during the maintenance phase [12,13,16,17].

### 4.3. Physical Activity

Appendix A presents IPAQ physical activity-related data. GLM multivariate analysis of selected IPAQ variables revealed Wilks’ Lambda time (*p* = 0.003) and group × time (*p* = 0.563) effects. Univariate analysis revealed significant time effects in vigorous (*p* = 0.002) and total (*p* = 0.003) physical activity effects with no significant group × time effects observed. However, pairwise comparisons revealed evidence that those in the CV group maintained more vigorous and total physical activity levels during the study. 

### 4.4. Weight and Body Composition

Appendix A provides a summary of body weight and body composition results. Wilks’ Lambda analysis revealed time (*p* < 0.001) and group × time interaction (*p* < 0.119) effects were revealed with GLM multivariate analysis of body weight, fat mass, lean mass, bone mineral content, and body fat. Univariate analysis revealed a significant interaction (*p =* 0.044) between groups in lean mass, with those in the CV group experiencing an increase in lean mass during the study while those in the WW group experienced a reduction in lean mass. Figure 4 presents mean changes with 95% CIs for body composition data. Participants in the CV group observed significantly greater fat mass and percent body fat loss and increases in lean mass during the intervention. After 16 weeks, weight loss was similar between groups (WW −6.1 [−7.8, −4.6], CV −4.9 [−3.2, −6.2] kg, *p* = 0.264). However, those in the CV group observed greater reductions in fat mass (WW −2.9 [−6.7, −0.2], CV −6.4 [−9.2, −3.6] kg, *p* = 0.081) and increases in lean mass (WW −2.5 [−4.3, −0.7], CV 1.3 [−0.6, 3.2] kg, *p* = 0.005) resulting in more favorable changes in percent body fat (WW −1.4 [−4.1, 1.2], CV −4.7 [−7.5, −1.8]%, *p* = 0.098). 

### 4.5. Fitness and Health-Related Variables

Appendix A presents fitness-related variables, while Appendix A presents health-related variables. Multivariate GLM analysis revealed Wilks’ Lambda time (*p* < 0.001) and group × time (*p* < 0.033) effects in fitness-related variables. Univariate analysis revealed significant time effects in peak VO_2_ (*p* < 0.001) and time to peak VO_2_ (*p* < 0.001), indicating both groups experienced improvements in aerobic capacity. Participants in the CV group experienced significantly greater improvements in bench press load-volume (*p* = 0.002). In terms of markers of health, time effects were observed in resting heart rate (*p* = 0.013), systolic blood pressure (*p* = 0.012), diastolic blood pressure (*p* = 0.005), waist circumference (*p* < 0.001), and hip circumference (*p* < 0.001), demonstrating both groups experienced improvements in these markers of health with no significant differences observed between groups.

### 4.6. Metabolic and Hormonal Profiles

Table 2 presents glucose homeostasis, leptin, and lipid profile data. Multivariate GLM analysis revealed overall Wilks’ Lambda time (*p* = 0.148) and group × time (*p* = 0.267) effects for glucose homeostasis variables. Neither intervention affected fasting glucose levels. Participants in the CV group experienced significant reductions in fasting insulin and HOMA levels over time. However, these values started and remained higher than those in the WW group. Analysis using changes from baseline as well as use of baseline values as a covariate were used to control for these differences and confirm results. Multivariate GLM analysis of leptin and lipid variables revealed a time (*p* < 0.001) and group × time (*p* = 0.098) effect for lipid-related variables. Univariate analysis revealed significant time effects in leptin (*p* < 0.001), total cholesterol (CHL: *p* < 0.001), low-density lipoproteins (LDLc: *p* < 0.010), and high-density lipoproteins (HDLc: *p* < 0.001) with a significant interaction observed in HDLc. Pairwise comparisons revealed that those in the CV group maintained or increased HDLc while those in the WW group experienced a slight decrease in HDLc. There was also evidence that those in the CV group experienced a greater reduction in the CHL–HDLc ratio and triglycerides (TG). 

## 5. Discussion and Conclusions

The aim of this this study was to compare the effects of adherence to two popular diet and exercise programs with different types of diet and/or exercise interventions on changes in weight, body composition, and markers of health and fitness in obese women. The WW intervention represented a standard diet-based intervention with weekly counseling sessions and recommended increases in physical activity, while the other program involved providing meal plans to help individuals adhere to energy and macronutrient intake guidelines with a supervised resistance-based circuit training and walking exercise intervention. We hypothesized that participating in a supervised exercise program that included resistance exercise while adhering to a higher protein diet would promote more favorable changes in body composition and markers of health and fitness than adherence to a standard diet-based intervention with an unsupervised recommendation to increase physical activity. Both interventions were effective in promoting weight loss; however, participants following a more structured diet and exercise intervention observed more favorable body composition, health, and fitness outcomes. The following provides additional analysis of the present findings.

### 5.1. Primary Outcome—Weight Loss and Body Composition

The primary aim of this clinical trial was to determine whether a menu-based higher protein low-fat diet combined with supervised resistance-based exercise training was more effective at promoting fat loss and improvements in health-related fitness outcomes than adherence to a diet-based program with weekly counseling and exercise encouragement. Results revealed that over time, both groups lost a similar amount of body weight. However, the CV intervention led to a greater decrease in fat mass and a greater increase in lean mass, resulting in a significant improvement in body fat percentage. Present findings agree with previous studies conducted by our group [10,11,12,13,14,15,16,18,19,20,21,22], which found that individuals participating in this program observed significant reductions in body mass, fat mass, BMI, and waist circumference while preserving lean body mass and resting energy expenditure. Additionally, that adherence to a more structured diet plan promoted less energy intake with a greater proportion of dietary protein intake [17]. They are also consistent with other investigations that found that adherence to a resistance-based circuit training with an energy-restricted high-protein diet resulted in greater loss of body fat and preservation of lean body mass [19,37,38,39]. Consistent with current findings, adherence to the WW program has been reported to promote reductions in body weight, fat mass, BMI, and waist circumference, though the preservation of lean body mass has not been demonstrated [40,41]. Both groups lost over 5% body weight in the 16-week period, which agrees with previous research evaluating adherence to these diets for 12 weeks to 12 months in duration [18,21,42,43,44,45,46,47,48,49,50,51]. This level of weight loss can be defined as clinically meaningful if maintained one year after treatment [52]. Modest weight loss can also lead to improvements in health markers, such as decreased blood pressure and increased HDLc, which should be the main target of clinicians, more so than the focus on scale weight [53,54].

### 5.2. Secondary Outcomes—Markers of Fitness and Health

As a secondary aim, this study assessed the changes in markers of fitness and health between the two diet programs, to include aerobic capacity, muscular strength, muscular endurance, resting hemodynamics, fasting glucose homeostasis, and lipid-related variables. The CV and WW groups both experienced fitness improvements such as increased aerobic capacity (peak VO_2_ and time to peak VO_2_) over the 16-week study period. Furthermore, the CV group experienced a significant improvement in bench press lifting load-volume, a measure of upper body muscular endurance. Neither group experienced changes in muscular strength nor lower body muscular endurance. Other trials from our lab evaluating changes in muscular strength and cardiovascular fitness following participation in the CV program have observed significant increases in upper and lower body 1RM strength [12,13,15,16,17,18,19,20,21]. In the present study, participants did not increase 1RM strength. However, it should be noted that participants in this study had greater upper and lower body 1RM strength at baseline compared to other trials, which may have influenced results. We are unaware of studies reporting the effects of adherence to the WW study on changes in strength, muscle endurance, and/or cardiovascular fitness.

Both interventions promoted health improvements that included a reduction in resting heart rate, systolic blood pressure, diastolic blood pressure, waist circumference, and hip circumference. Prior studies have reported reductions in blood pressure when following the CV program [12,17,20,21] and the WW program [43]. While the CV program emphasized higher protein intake, previous research by Witjaksono and colleagues [55] demonstrated that when combined with a reduced caloric intake for eight weeks, the proportion of protein did not significantly affect waist circumference without exercise included. As mentioned above, these modest health improvements are likely a result of the ~5% weight loss experienced in each group [54]. While neither intervention affected fasting glucose levels, the CV participants experienced significant reductions in fasting insulin and HOMA levels throughout the study, similar to our previous investigations utilizing a hypocaloric high-protein diet with resistance circuit training [12,13,15,18]. One study from our lab reported a significant reduction in HOMA with both the CV and WW over 12 weeks [10].

In our investigation, both the CV and WW groups experienced reductions in leptin, total cholesterol, and LDLc, and the WW group experienced a decrease in HDLc, while the CV group maintained or increased HDLc. There was also evidence that those in the CV group experienced a greater reduction in the CHL–HDLc ratio and TG. In a recent review regarding the effects of exercise on HDL, Ruiz-Ramie et al. [56] reported that it is likely that both exercise dose-threshold and exercise intensity-threshold must be achieved in order to sufficiently increase HDLc [56]. Therefore, it seems likely that the greater frequency and intensity of exercise in the CV program accounted for the observed improvement in HDLc. Previous research from our lab has also shown improvement or maintenance in HDLc while following a reduced-calorie diet, no matter the macronutrient composition when incorporating the resistance-based circuit-training program [10,12,18]. Other investigations have also shown a superior effect of elevated protein ingestion [18,20,37,38,39,57] to reduce triglycerides while maintaining HDLc. These changes have been attributed to replacing dietary carbohydrate with protein thereby enhancing the efficiency of metabolizing fat particularly when dietary fat intake is maintained or reduced [58,59]. Previous investigations of the WW program have shown varying lipid panel changes following program intervention. Bellisle et al. [43] observed reductions in CHL and LDLc with no changes in HDLc, TG, or in CHL–HDLc ratio after three months of program participation. Morgan and associates [60] observed decreased TG, LDLc, and HDLc following six months of participation. Moreover, Dansinger and colleagues [40] observed a reduction in LDLc, increase in HDLc, and 10% decrease in LDLc–HDLc ratio after one year with no alteration in TG. In contrast, Jebb and coworkers [46] observed no significant changes in lipid panel values following 12 months of participation in the WW program. Our findings support previous reports that adherence to a meal plan-based diet while engaged in a resistance-based circuit training program can promote reductions in serum leptin levels [10,12,18,61]. Further, research from our lab has evaluated the reduction in metabolic syndrome (MetS) prevalence when following commercially available weight loss programs such as Curves and Weight Watchers for 12 weeks, reporting that the addition of the structured exercise program in Curves is more efficacious at reducing the prevalence of MetS [10]. We have also assessed the macronutrient distribution of the CV reduced-calorie diet protocol and found each to be equally effective at reducing the prevalence of MetS when combined with the structured exercise program [22].

Finally, the cost-effectiveness of these diet and exercise interventions should be considered. While the WW group lost more weight and fat mass per dollar spent, they also lost more fat-free mass, compared to the CV group who lost weight and fat mass while preserving fat-free mass, therefore resulting in a greater improvement in body fat percentage per dollar spent [62]. However, previous researchers have found Weight Watchers the most cost-effective commercial weight loss program [63,64]. For example, Johnson and colleagues [47] reported that the WW program was five times more cost-effective than Curves, yet they also stated that the CV program was more intense, and the participants lost 5% body weight in half the time as the WW participants. For this reason, the CV program has been included as an option in insurance-reimbursement programs. Similarly, the WW program has been offered in federally funded initiatives for low-income groups to prevent, treat, and manage chronic diseases [47,65]. Thus, the WW has been proven both effective for weight loss and cost-effective in comparison to several other commercial weight loss programs [9,40,41,60,63] and has even been suggested as a program to which clinicians should refer overweight clients [66]. This investigation promotes the argument that the CV program should also be considered effective for clinicians and federal programs to promote weight loss and health improvement; in fact, both programs have been labeled as approved weight-loss interventions by the CDC WISEWOMAN program [47].

### 5.3. Limitations

There are several limitations in conducting weight-loss clinical trials that should be considered when interpreting the results of this investigation. First, participants who volunteer to participate in weight-loss trials are often more motivated to adhere to weight loss programs than the general population. During this study, participants were monitored and encouraged to meet program eligibility requirements. These efforts resulted in 81% compliance for exercise sessions in the CV group and 90% compliance in weekly meeting attendance in the WW group. Although these are similar to compliance rates observed in other long-term weight loss trials, results may not generalize to populations that do not receive similar monitoring and support. Second, despite the provision of meal plans and weekly compliance checks, participants in the CV group did not fully meet intended energy and macronutrient intake, particularly during phase 3 of the diet in which individuals were to consume an energy-balanced diet. Whether this detriment was due to a lack of compliance or challenges in recording food intake remains to be determined, but this observation is consistent with our other studies. Third, the programs evaluated in this study cost money to participate. Therefore, these programs may not be affordable to some populations unless they receive reimbursement. Fourth, the results observed in this trial are limited to the population studied, women who are sedentary and obese, and might not apply to other populations. Fifth, this trial only lasted four months. Therefore, it is unclear whether weight loss and/or positive changes in body composition, fitness, and health can be maintained. Finally, results are always limited to the challenges of conducting clinical trials in overweight populations, particularly when requiring adherence to diet and exercise.

### 5.4. Conclusions

Overall, both programs proved effective at decreasing body weight and improving some markers of health and fitness. The CV program promoted greater decreases in fat mass and percent body fat, a reduction in fasting insulin and HOMA levels, and an increase in lean mass and upper body muscular endurance. These findings support contentions that following a more structured diet plan, increasing the proportion of dietary protein, and incorporating a supervised resistance-exercise program can be an effective way to maintain fat-free mass and resting energy expenditure during a hypoenergetic diet intervention. These findings can be helpful to individuals and clinicians in understanding that different diet and exercise interventions may promote differential effects on body composition and health outcomes. Additional research should aim to understand the long-term effects of different diet and exercise interventions on weight loss, body composition, and health markers.

## Figures and Tables

**Figure 1 ijerph-19-04877-f001:**
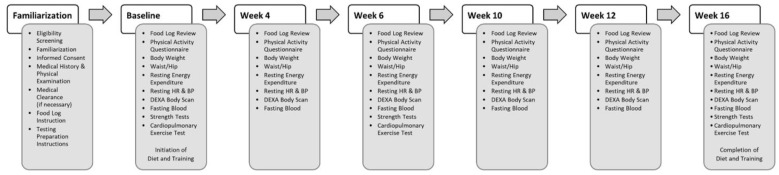
Overview of testing sessions and assessments performed. DEXA is dual-energy X-ray absorptiometry, HR is heart rated, BP is blood pressure.

**Figure 2 ijerph-19-04877-f002:**
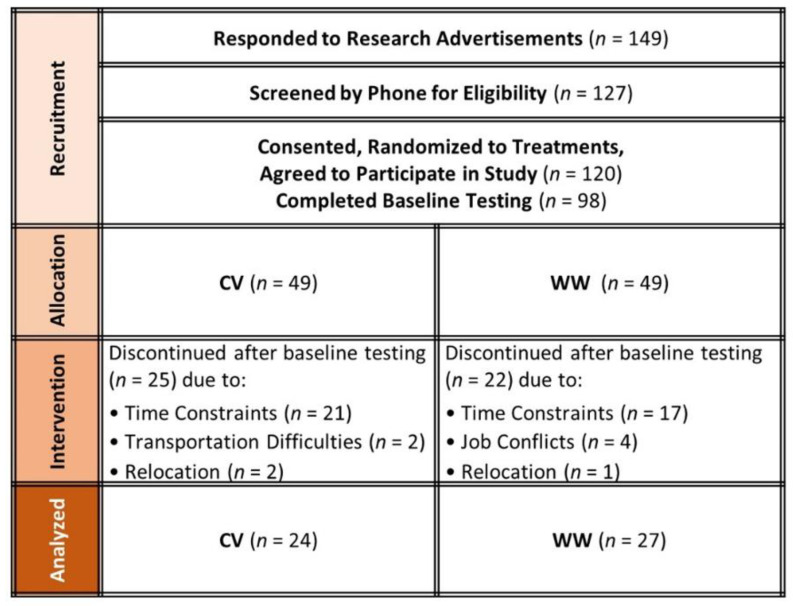
CONSORT diagram. CV represents the Curves fitness and weight management program. WW represents the Weight Watchers^®^ Momentum™ intervention group.

**Figure 3 ijerph-19-04877-f003:**
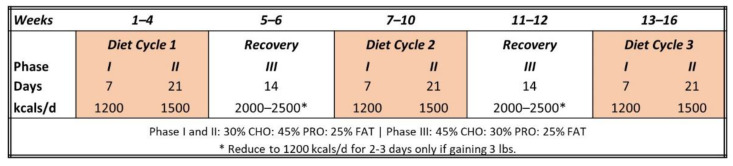
Overview of the CV diet intervention.

**Figure 4 ijerph-19-04877-f004:**
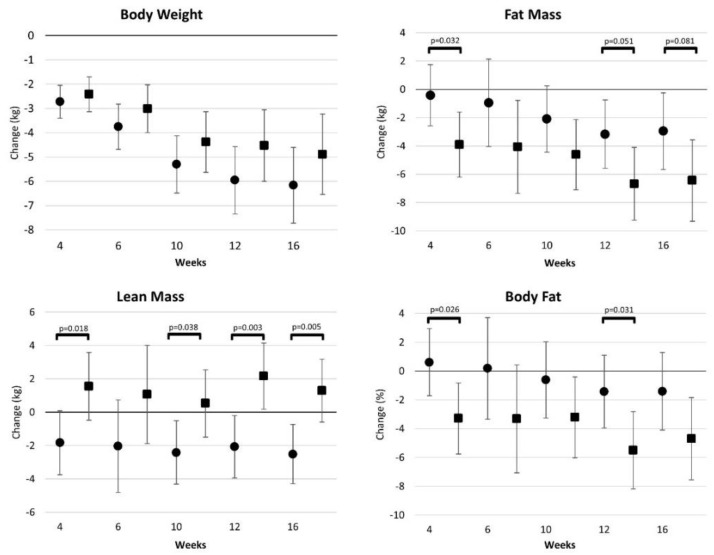
Body composition changes from baseline (mean [95% CI]) observed for Weight Watchers (●) and Curves (■) interventions.

**Table 1 ijerph-19-04877-t001:** Baseline Characteristics.

	WW	CV	Mean	*p*-Level
	(*n* = 27)	(*n* = 24)	(*n* = 51)	
Age (years)	34.0 ± 6.1	35.1 ± 9.2	34.5 ± 7.7	0.633
Weight (kg)	89.5 ± 13.6	90.8 ± 15.6	90.1 ± 14.5	0.770
Height (cm)	162.6 ± 6.7	163.0 ± 7.4	162.8 ± 7.0	0.843
BMI (kg/m^2^)	33.9 ± 5.2	34.1 ± 5.1	34.0 ± 5.1	0.899
Body fat (%)	45.1 ± 7.3	48.1 ± 6.5	46.5 ± 7.1	0.138

Data are means and ± standard deviations for Curves (CV) and Weight Watchers (WW) groups.

**Table 2 ijerph-19-04877-t002:** Metabolic and Hormone Profiles.

Weeks										
	0		4	6	10	12	16	Effect	*p*-Level	η_p_^2^
Glucose (mg/dL)	WW	89.1 ± 9	89.2 ± 8	85.3 ± 8	87.4 ± 6	88.1 ± 5	89.0 ± 7	Group	0.658	0.004
	CV	91.5 ± 15	88.7 ± 14	86.9 ± 9	90.7 ± 19	88.4 ± 8	87.9 ± 16	Time	0.237	0.028
								G × T	0.628	0.012
Insulin (IU/mL)	WW	11.0 ± 5	10.7 ± 4	10.7 ± 8	11.2 ± 6	11.1 ± 5	10.6 ± 5	Group	0.000	0.307
	CV	20.5 ± 12	15.9 ± 7 ^†^	16.8 ± 8	14.5 ± 7 ^†^	17.4 ± 7	16.1 ± 8 ^†^	Time	0.214	0.029
								G × T	0.240	0.028
HOMA	WW	2.40 ± 1	2.35 ± 1	2.27 ± 2	2.39 ± 1	2.42 ± 1	2.35 ± 1	Group	0.000	0.310
	CV	4.71 ± 3	3.44 ± 1 ^†^	3.66 ± 2	3.23 ± 2 ^†^	3.77 ± 1 ^†^	3.56 ± 2 ^†^	Time	0.134	0.036
								G × T	0.174	0.032
Leptin (ng/mL)	WW	51.3 ± 22	46.8 ± 27 ^†^	43.2 ± 32	45.5 ± 35 ^†^	45.5 ± 30	42.3 ± 29 ^†^	Group	0.837	0.001
	CV	66.3 ± 27 ^b^	53.0 ± 26	50.0 ± 22	42.1 ± 27	48.3 ± 23	52.7 ± 25	Time	0.001	0.094
								G × T	0.123	0.036
Total Cholesterol (mg/dL)	WW	186 ± 30	172 ± 27 ^†^	172 ± 24 ^†^	175 ± 31 ^†^	177 ± 29	173 ± 30 ^†^	Group	0.328	0.020
	CV	175 ± 30	164 ± 31 ^†^	168 ± 33	167 ± 31	173 ± 30	171 ± 33	Time	0.001	0.091
								G × T	0.589	0.015
LDLc (mg/dL)	WW	110 ± 26 ^b^	103 ± 24 ^†^	102 ± 23 ^†^	105 ± 25	101 ± 24 ^†^	96 ± 22 ^†^	Group	0.837	0.001
	CV	97.7 ± 26	90.5 ± 29	93.3 ± 28	94.0 ± 25	97.0 ± 27	92.9 ± 28	Time	0.010	0.062
								G × T	0.260	0.026
HDLc (mg/dL)	WW	52.1 ± 14	46.8 ± 13 ^†^	46.3 ± 10 ^†^	46.4 ± 12 ^†^	48.0 ± 11 ^†^	50.8 ± 14	Group	0.699	0.003
	CV	51.7 ± 15	49.7 ± 12	51.7 ± 13	52.7 ± 14 ^b^	52.3 ± 14	54.3 ± 16	Time	0.000	0.115
								G × T	0.017	0.059
CHL:HDL Ratio	WW	3.77 ± 1.0	3.87 ± 1.0	3.88 ± 1.0 ^b^	3.96 ± 1.1 ^†^	3.94 ± 1.3	3.64 ± 1.2	Group	0.959	0.000
	CV	3.60 ± 1.1	3.48 ± 0.9	3.42 ± 1.0	3.42 ± 0.9 ^b^	3.55 ± 1.0	3.45 ± 1.0	Time	0.214	0.029
								G × T	0.139	0.034
Triglycerides (mg/dL)	WW	120 ± 44	122 ± 49	115 ± 35	122 ± 53	123 ± 49	131 ± 99	Group	0.505	0.009
	CV	126 ± 59	113 ± 58	115 ± 50	109 ± 49 ^†^	119 ± 54	124 ± 63	Time	0.363	0.021
								G × T	0.678	0.010

Data are expressed as means ± standard deviations for the Curves (CV) and Weight Watchers (WW) groups. General Linear Model analysis revealed overall Wilks’ Lambda Time (*p* = 0.148) and Group × Time (*p* = 0.267) effects for glucose homeostasis variables and Time (*p* < 0.001) and Group × Time (*p* = 0.098) for lipid related variables. Greenhouse-Geisser univariate *p*-levels are listed for Group (G), Time (T), Group × Time (G × T). HOMA = homeostasis model assessment, LDLc = low density lipoprotein cholesterol, HDLc = high density lipoprotein cholesterol, CHL = cholesterol. Partial ETA squared (η_p_^2^) are presented to assess effect sizes where 0.01 is considered small, 0.09 considered medium, and 0.25 considered large effect sizes. Superscripts † = *p* < 0.05 difference from baseline value; a = *p* < 0.05 difference between CV and WW groups; and, ^b^ = *p* > 0.05 to *p* < 0.10 difference between CV and WW groups.

## Data Availability

Data and/or statistical analyses are available upon request on a case-by-case basis for non-commercial scientific inquiry and/or educational use as long as IRB restrictions and research agreement terms are not violated.

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
