# Peer review of "Comparison of Two Diet and Exercise Approaches on Weight Loss and Health Outcomes in Obese Women"

_ijerph, 2022, doi:10.3390/ijerph19084877_

Round 1

Reviewer 1 Report

In the present study, authors investigated the impact of two diet and exercise approaches on obese women. The whole study was well designed and conducted. All the data supports the conclusion well. It was found that CV training program is more effective in terms of fat loss, lean mass and blood lipids. The result of the present study is meaningful to choose and even explain the effective weight control program. Some suggestions are listed below.

  1. It is suggested that more discussion should be added to explain why CV is more effective than WW;
  2. In table 2, it is noted that insulin and HOMA value in CV group are higher than that of WW group. Certain value in CV even presents nearly twice of WW. What is the main reason and how this impact on the result?

Author Response

Reviewer 1

Comment:

In the present study, authors investigated the impact of two diet and exercise approaches on obese women. The whole study was well designed and conducted. All the data supports the conclusion well. It was found that CV training program is more effective in terms of fat loss, lean mass and blood lipids. The result of the present study is meaningful to choose and even explain the effective weight control program. Some suggestions are listed below.

Response:

Thank you for your review of this paper.  We have made changes as requested.

Comment:

  1. It is suggested that more discussion should be added to explain why CV is more effective than WW;

Response:

We hypothesized that participating in a supervised exercise program that included resistance-exercise while adhering to a higher protein diet would promote more favorable changes in body composition and markers of health and fitness than adherence to a standared diet-based intervention with an unsupervised recommendation to increase physical activity. We found that both interventions were effective in promoting weight loss; however, participants following a more structured diet and exercise intervention observed more favorable body composition, health, and fitness outcomes.  As we have previously reported, we feel this is due to increasing protein intake during a hypenergetic diet combined with resistance-exercise that helps maintain fat free mass and prevent reductions in resting energy expenditure. Additionally, adherence to meal plans and supervised exercise helps participant compliance. We emphasized this to a greater degree in the conclusions.

Comment:

  1. In table 2, it is noted that insulin and HOMA value in CV group are higher than that of WW group. Certain value in CV even presents nearly twice of WW. What is the main reason and how this impact on the result?

Response:

Thank you. It is unclear why insulin and HOMA levels were higher in this randomized cohort despite match based body mass and BMI.  However, we controlled for this by assessing mean changes from baseline with 95% CI’s as well as using baseline values as a covariate to confirm results.  We added a description about this in the results.  

Reviewer 2 Report

Reviewer comments and suggestions

The current study compared the efficacy of two popular weight loss approaches on weight loss, body composition, and markers of health in sedentary obese women. The study included 51 sedentary women (age 34.5±7.7yrs; weight 90.0±14.5 kg; BMI 34.0±5.1 kg/m2) were matched and randomized to join the Weight Watchers® Momentum™ (WW) or Curves® (CV) Fitness and Weight Management program for 16-wks. 

The WW group (n = 27) was provided a point-based diet program, received weekly progress checks, and was encouraged to exercise. Participants in the CV group (n = 24) followed a menu-based higher protein/low fat diet (1,200 kcal/d) for 1-wk; 1,500 kcal/d diet for 3-wks; and 2,000-2,500 kcals/d for 2-wks that was repeated three times (except last segment).

After 16 weeks, both groups lost weight (WW -6.1 [-7.8, -4.6], CV -4.9 [-6.2, -3.2] kg, p = 0.264). Participants in the CV group observed greater reductions in fat mass (WW -2.9 [-6.7, -0.2], CV -6.4 [-9.2, -3.6] kg, p = 0.081) and increases in lean mass (WW -2.5 [-4.3, -0.7], CV 1.3 [-0.6, 3.2] kg, p = 0.005) resulting in more favorable changes in percent body fat (WW -1.4 [-4.1, 1.2], CV -4.7 [-7.5, -1.8] %, p = 0.098). Both groups observed improvements in peak aerobic capacity and muscular endurance and in the CV group, they experienced a greater increase in HDLc and a reduction in the CHL: HDLc ratio and triglycerides. Finally, the study concluded that both interventions promoted weight loss and improvements in fitness and markers of health. 

The paper has nicely complied and studied well in terms of scientific quality. However, in many places, the authors need to do a few small corrections to the manuscript. Based on my view, below are the comments that need to be incorporated into the revised version of the manuscript.

  1. Line no 58-60 need a reference for this
  2. Line 71-72 is already discussed above. please delete the lines
  3. What does it mean,” manipulation of macro-nutrient targets likely plays”
  4. Please avoid long sentence check lines from 77-92 and 150-155
  5. Line 216 Previous studies needed more references to valid the sentence information
  6. Line 297-300 The study needs to be discussed or do not put many references
  7. In line 355 a typo error was present.
  8. I have seen that the authors highlight their previous studies too much in the manuscript that was not necessarily required. Please check the introduction and discussion for these issues.
  9. Line 388 What does it mean? “maintenance ranging from 12- weeks to 12-months [15,17,42-53]”
  10. Line 439-440 is there was any specific reason for this
  11. Please check the references and modify them based on the journal guidelines. Reference number 9, 11, 17,32, 38,39,43,49,50,59,61 and 63

Author Response

Reviewer 2

Reviewer comments and suggestions

The paper has nicely complied and studied well in terms of scientific quality. However, in many places, the authors need to do a few small corrections to the manuscript. Based on my view, below are the comments that need to be incorporated into the revised version of the manuscript.

Response:  Thank you for your review and supportive comments.  We have tried to address comments as noted below.

  1. Line no 58-60 need a reference for this

Response. Thank you. References added.

  1. Line 71-72 is already discussed above. please delete the lines

Response  Thank you, removed.

  1. What does it mean,” manipulation of macro-nutrient targets likely plays”

Response:  Thank you. Clarified

  1. Please avoid long sentence check lines from 77-92 and 150-155

Response:  Shortened a bit as requested.

  1. Line 216 Previous studies needed more references to valid the sentence information

Reponse:  Rephrased. 

  1. Line 297-300 The study needs to be discussed or do not put many references

Response:  Reduced number of references supporting this statement as requested.

  1. In line 355 a typo error was present.

Response:  Thank you.

  1. I have seen that the authors highlight their previous studies too much in the manuscript that was not necessarily required. Please check the introduction and discussion for these issues.

Response:  This study is one of a series of studies from a women’s health and fitness research initiative that investigated the role of adherence to a higher protein diet with resistance exercise as a means of promoting more optimal weight loss and health outcomes in obese women. There is a logical progression of intervention studies conducted that led to this study. It’s important that readers know the context that this study was one of a series our group has conducted over the years and they can easily find those related references so they can compare results. We feel this is important in this manuscript.  Nevertheless, we evaluated each listing and narrowed down where we thought was appropriate.

  1. Line 388 What does it mean? “maintenance ranging from 12- weeks to 12-months [15,17,42-53]”

Response: Clarified

  1. Line 439-440 is there was any specific reason for this

Response:  Yes, replacing dietary carbohydrate with protein has been consistently shown to reduce triglycerides. These changes have been attributed to a greater reliance on fat metabolism and thereby more efficient use of triglycerides as fuel particularly when the diet does not increase fat intake. We added a statement to this effect and a couple of references.

  1. Please check the references and modify them based on the journal guidelines. Reference number 9, 11, 17,32, 38,39,43,49,50,59,61 and 63

Response:  Revised.

Thank you!